# Physical Rehabilitation Needs in the BRICS Nations from 1990 to 2017: Cross-National Analyses Using Data from the Global Burden of Disease Study

**DOI:** 10.3390/ijerph17114139

**Published:** 2020-06-10

**Authors:** Tiago S. Jesus, Michel D. Landry, Helen Hoenig, Yi Zeng, Sureshkumar Kamalakannan, Raquel R. Britto, Nana Pogosova, Olga Sokolova, Karen Grimmer, Quinette A. Louw

**Affiliations:** 1Global Health and Tropical Medicine (GHTM) & WHO Collaborating Centre for Health Workforce Policy and Planning, Institute of Hygiene and Tropical Medicine - NOVA University of Lisbon (IHMT-UNL), Rua da Junqueira 100, 1349-008 Lisbon, Portugal; 2School of Medicine, Duke University, Durham, NC 27710, USA; mike.landry@duke.edu; 3Duke Global Health Institute (DGHI), Duke University, Durham, NC 27710, USA; 4Physical Medicine and Rehabilitation Service, Durham Veterans Administration Medical Center, Durham, NC 27705, USA; helen.hoenig@va.gov; 5Division of Geriatrics, Department of Medicine, Duke University Medical Center, Durham, NC 27710, USA; 6Center for Study of Aging and Human Development and Geriatrics Division, School of Medicine, Duke University, Durham, NC 27710, USA; zengyi@nsd.pku.edu.cn; 7National School of Development and Raissun Institute for Advanced Studies, Peking University, Beijing 100871, China; 8Public Health Foundation of India (PHFI), South Asia Centre for Disability Inclusive Development and Research (SACDIR), Indian Institute of Public Health, Hyderabad 500 033, (IIPH-H), India; Suresh.Kumar@lshtm.ac.uk; 9Rehabilitation Science Post Graduation Programs of Universidade Federal de Minas Gerais and Universidade Federal de Juiz de Fora, Juiz de Fora 36036-900, Brazil; r3britto@gmail.com; 10National Medical Research Center of Cardiology, Moscow 524901, Russian Federation; nanapogosova@gmail.com (N.P.); birdname@gmail.com (O.S.); 11Department of Health and Rehabilitation Sciences, Physiotherapy Division, Stellenbosch University, Stellenbosch 7505, South Africa; ubiquitous598@hotmail.com (K.G.); qalouw@sun.ac.za (Q.A.L.)

**Keywords:** global burden of disease, global health, health services needs and demand, BRICS, rehabilitation

## Abstract

*Background*: This study analyzes the current and evolving physical rehabilitation needs of BRICS nations (Brazil, Russian Federation, India, China, South Africa), a coalition of large emergent economies increasingly important for global health. *Methods:* Secondary, cross-national analyses of data on Years Lived with Disability (YLDs) were extracted from the Global Burden of Disease Study 2017. Total physical rehabilitation needs, and those stratified per major condition groups are analyzed for the year 2017 (current needs), and for every year since 1990 (evolution over time). ANOVAs are used to detect significant yearly changes. *Results*: Total physical rehabilitation needs have increased significantly from 1990 to 2017 in each of the BRICS nations, in every metric analyzed (YLD Counts, YLDs per 100,000 people, and percentage of YLDs relevant to physical rehabilitation; all *p* < 0.01). Musculoskeletal & pain conditions were leading cause of physical rehabilitation needs across the BRICS nations but to varying degrees: from 36% in South Africa to 60% in Brazil. Country-specific trends include: 25% of South African needs were from HIV-related conditions (no other BRICS nation had more than 1%); India had both absolute and relative growths of pediatric rehabilitation needs (*p* < 0.01); China had an exponential growth in the per-capita needs from neurological and neoplastic conditions (*p* < 0.01; r^2^ = 0.97); Brazil had a both absolute and relative growth of needs coming from musculoskeletal & pain conditions (*p* < 0.01); and the Russian Federation had the highest neurological rehabilitation needs per capita in 2017 (over than three times those of India, South Africa or Brazil). *Conclusions*: total physical rehabilitation needs have been increasing in each of the BRICS nations, both in absolute and relative values. Apart from the common growing trend, each of the BRICS nations had own patterns for the amount, typology, and evolution of their physical rehabilitation needs, which must be taken into account while planning for health and physical rehabilitation programs, policies and resources.

## 1. Introduction

The BRICS countries (i.e., Brazil, Russian Federation–called Russia hereafter, India, China, South Africa) are increasingly recognized as important players in global health and development [1,2,3,4,5,6,7,8,9]. Traditionally, the G7 (the group of the seven most powerful world economies) steered major health initiatives globally, through policies, priorities and developmental aid to support and improve health in Low-and Middle-Income Countries (LMICs) [1]. However, global health attention has been turning to the strategic role of emerging economies, especially the BRICS nations. These are the five large emerging economies that formalized a coalition and agenda for economic growth and health gains apart from the traditional, western global agenda [1,2,4,5,6,7,8,9].

The BRICS countries, which formalized their coalition in 2006, generate 25% of the world’s gross national income, have approximately 40% of the world’s population, approximately 50% of the world’s poor, and represent 40% of the global burden of disease [2]. Through strategic cooperation and inter-BRICS policies, the BRICS nations increasingly seek to translate their economic growths into improved population health [7,10,11]. Their health ministries have been met annually to discuss synergies, priorities and innovations tailored to their resource-constrained settings [7,10]. The BRICS nations’ agendas have been different than the Western nations, with a particular emphasis on social justice and equity in health in context of their unique, multifaceted health challenges [7,10]. The BRICS’ national health challenges include important increases in the prevalence of non-communicable, chronic diseases [3,12,13,14,15,16,17,18] along with a still prevalent burden of communicable conditions [19,20], multifaceted social determinants of health, and high inequalities in health and healthcare access [2,21]. As these challenges are similar to those of other LMICs, advances in the BRICS health policies, cooperation, and healthcare delivery have been inspiring for other countries with developing economies [4,6,10]. Finally, the BRICS countries have been providing concrete assistance to the LMICs; [7,8,9] for example, through a “South-South Cooperation” over 55 years, China has dispatched medical teams, constructed facilities, distributed drugs and medical devices, and has trained local health workers in more than 66 countries in need [9].

Despite their economic developments and coalition, the BRICS countries face themselves a shortage of key health resources (e.g., health and allied health professionals) for their growing and complex health needs, in the context of many other societal demands [21,22]. These needs accentuate the complexity of planning for equitable and effective health and social care amidst rapid demographic, economic and epidemiological transitioning [3,12,13,14,15,16,17,18]. BRICS countries therefore had to be innovative to re-engineer the health and social care systems challenges (often limited healthcare finances, workforce, training, service planning and administration to address population healthcare needs), as well as the growing numbers of persons living with disability [23,24,25,26].

Worldwide and especially in emerging economies, increasing numbers of people now live with functional limitations [1,12,26]. This can be explained from the demographic and epidemiological transitions with increasing life expectancy, an ageing population and the subsequent burden of chronic diseases [18,26,27,28,29,30,31]. Not only are many people now living with chronic communicable diseases which previously were fatal, but there is also an increasing prevalence of non-communicable diseases which are lifestyle-related and/or come as complex, multiple co-morbidities resulting in varying types and degrees of long-term disabilities [1,13,32,33,34,35,36,37,38,39,40,41]. Rehabilitation is required to attenuate the effects of disability and optimize functioning in people with functional limitations from any health condition [25,42]. Failure to address individuals’ rehabilitation needs impacts on human functioning, social justice, human rights, productivity, long-term costs of care, and even could impact countries’ economic growth [42,43,44,45,46].

In line with increasing disability prevalence, a recent study using data from Global Burden of Disease Study 2017 found a 17% increase in the world’s physical rehabilitation needs per capita since 1990, and an almost twofold greater increase (29.9%) in upper-middle-income countries (UMICs), which include four of the BRICS countries (except India) [25]. The World Health Organization’s (WHO) Rehabilitation 2030 initiative advocates for the inclusion of rehabilitation in universal health coverage, across countries of all income levels [42]. All BRICS countries have now committed to universal health coverage, although with varied levels of coverage, principles and roll-out over the next decade [6,11,22,47]. It is therefore timely to determine the need for rehabilitation in BRICS countries. This information will not only support advocacy and strategic planning for the widespread inclusion of rehabilitation in the roll-out and expansion of universal health coverage in BRICS countries, but it will inform improvements in the planning for rehabilitation services in other LMICs. 

This paper aims to analyze the current and evolving physical rehabilitation needs of the BRICS countries. The specific study questions are:(1)How large are the physical rehabilitation needs in 2017 for each BRICS country (e.g., in nominal values, population-adjusted rates, age-standardized rates), and how have those values evolved since 1990?(2)Which condition groups (e.g., musculoskeletal, neurological, cardiothoracic) account for the highest rates of physical rehabilitation need for each of the BRICS countries in 2017, and how have those values evolved since 1990?

## 2. Materials and Methods 

This paper refers to a secondary, cross-national comparative analysis of global epidemiological data in the public domain. To estimate the physical rehabilitation needs for each of the BRICS nations, we use data from the 2017 Global Burden of Disease Study (GBD) [48]. Specifically, we combine the methods of two recent papers using GBD data to analyzing global physical rehabilitation needs [25,49]. The first uses GBD data to determining total physical rehabilitation needs, i.e., for all conditions relevant to physical rehabilitation combined [25]. The second stratifies these needs by condition type, e.g., musculoskeletal, neurological, cardiothoracic [49]. The use of those standard methods allows for the new findings for the BRCIS nations to be compared with the existing global benchmarks, i.e., physical rehabilitation needs for the world and for the groups of countries for all income levels [25,49].

To determine the physical rehabilitation needs for the BRICS nations, we apply and combine the abovementioned standard methods as follows: In April 2019, public-domain data from the GBD 2017 were systematically extracted from a freely-available web platform: the Global Health Data Exchange tool (http://ghdx.healthdata.org/gbd-results-tool).

With the due measures to avoid double counting [25], data were extracted for the set of health conditions likely benefiting from physical rehabilitation. Previously, these were systematically determined and tested for robustness (i.e., similar patterns of results were found for a sub-set of conditions) [25]. Table 1, left column, details the set of conditions deemed as likely benefiting from physical rehabilitation.

Among the GBD “measures”, we extract data only for Years Lived with Disability (YLDs), due its exclusive focus on non-fatal health losses. YLDs consist of the years lived with any short-term or long-term health loss weighted for severity by disability weights. For stroke, for example, disability weights vary from 0.019 for mild consequences to 0.588 for severe consequences plus cognition problems. Details on how YLDs and disability weights are determined, and the disability weights for all conditions, are available elsewhere [48,50].

For “years”, data were extracted for every year between 1990 and 2017, for a more precise determination of the evolving trend. For “location”, YLDs were extracted at the national level for each of the BRICS. No sub-national data were extracted, as we focused on nation-wide and cross-national analyses.

As for “metrics”, we extracted YLDs data for prevalent *number* (i.e., YLD counts), *rate* (i.e., YLDs per 100,000 people), and *percentage* (i.e., percentage of YLDs from the selected conditions relative to the total amount of YLDs).

Regarding “age”, we extracted YLDs both for *all ages* and *age-standardized* rates, the latter used to determine *age-standardized YLD Rates* (i.e., physical rehabilitation needs adjusted for both population size and ageing).

All the selected data were imported from the webtool to Excel spreadsheets for data storage, management, and analysis.

In the Excel spreadsheets, we summed YLDs within each of the five “locations”, four “metrics”, and 28 “years”, computed any percent changes from 1990 to 2017, plotted the entire time series [1990–2017] of the combined YLD values, and then determined which type of simple regression model (i.e., linear, exponential, or logarithmic) best fit the plotted data. We used visualization and *r*^2^ values for that. Given negligible differences in *r*^2^ values (<0.02 between the models), we retained the linear regression option.

To assess yearly changes of YLDs between 1990 and 2017, ANOVA was applied. This test considers the data on every year between 1990 and 2017, which increases preciseness. The significance level for the analysis was set at two subsequent levels: *p* = 0.05 and *p* = 0.01, the latter accounts for a Bonferroni correction (0.05/5 = 0.01) considering the analyses are made for five countries within each item/metric under study. The respective confidence intervals (CIs) in turn were used to analyze whether yearly changes for each BRICS nation significantly differed (i.e., did not overlap) from those of the 4 other BRICS countries or from the global benchmarks that we extracted from the literature–as the same methods were used [25,49].

Finally, using the analytical procedures above, we performed a subgroup analysis on the physical rehabilitation needs, stratified per six major groups of conditions–detailed in the Table 1′s right column. For that analysis, we only use *YLD Rates* as a metric, either through actual *YLD Rates* or through those transformed into percent values, e.g., percentage of the YLD Rates related to physical rehabilitation that came specifically from neurological conditions.

## 3. Results

We provide below the results for the two study questions:

### 3.1. Total Physical Rehabilitation Needs

Table 2 shows a significant increase in total physical rehabilitation needs from 1990 to 2017 across the five countries in all the metrics analyzed (*p* < 0.01); the exception being Age-Standardized YLD Rates for both India and Russia, with no significant changes since 1990 (*p* > 0.05). 

Per metric, Table 2 (see Appendix A, pages 1 to 4, for a visual representation of the data) shows the following trends: 

In YLD Counts (i.e., absolute YLD values), South Africa more than doubled their physical rehabilitation needs from 1990 to 2017 (i.e., 114.1% growth), similarly to low-income countries (i.e., 111.5% growth). India, Brazil, and China had a 91.6%, 82.6%, and 68.1% growth, respectively. Russia, in turn, had the lowest percentage growth in YLD counts (7.6%), substantially lower than any global benchmark: i.e., the lowest being 37.4% for high-income countries.

In YLD Rates (i.e., YLDs per 100,000 people), Russia had the highest value in 2017 (6393), but South Africa and China had the highest yearly growths (99% CIs: 56.8−90.0 and 47.6−60.9, respectively), each of them significantly higher (i.e., greater, non-overlapping 99% CIs) than those of the 3 other BRICS nations or any global benchmark.

In Age-standardized YLD Rates, (i.e., YLDs adjusted for both population size and ageing), South Africa had the highest value in 2017 (5131), and significantly greater yearly increases (i.e., non-overlapping 99% CIs) than all comparators; for example high-income countries had a significant decrease. Russia and India had non-significant yearly changes in this metric.

Finally, in the percentage of YLDs likely benefiting from physical rehabilitation among total YLDs, Brazil stands out with nearly two-thirds of their YLDs coming from rehabilitation-sensitive conditions in 2017 (66.2%). For the other BRICS countries or global benchmarks, values were all below 50%; in India little more than one-third (35.9%). In South Africa, the yearly growths in the YLDs percentage were significantly greater while in Russia significantly lower than in any other BRICS nations. Indeed, the 99% CIs of the Russia’s yearly growth was rather aligned (i.e., partly overlapping) with that of high-income nations. 

### 3.2. Needs by Condition Types

Table 3 (see Appendix A, pages 1 to 6, for a visual representation of the data) shows significant increases in rehabilitation-sensitive *YLD Rates* from 1990 to 2017 across the BRICS countries, for all the condition groups (*p* < 0.01). The exceptions are the South African’s YLD rates coming from pediatric and from neoplasm conditions (99% CIs: −1.60−1.53 and −0.06−0.62, respectively), although the latter had a significant increase within the 95% CI (0.03−0.53). 

The highest yearly increases in *YLD Rates* came from: (1) HIV-related conditions in South Africa (*b* = 63.8), yet with a logarithmic growth (i.e., greater growth rate in the earlier years); (2) musculoskeletal & pain conditions in both Brazil and China (*b* = 25.4 and *b* = 25.0, respectively); (3) neurological conditions in China (*b* = 15.6), within exponential type of growth (i.e., greater growth rate in the more recent years), and (4) pediatric conditions in India (*b* = 7.18). In each of these cases, the growths were significantly greater (i.e., higher, non-overlapping 95% CIs) than those of any comparators for the same condition group. Finally, Russia stands out with the highest YLD Rates for neurological conditions in 2017, e.g., over than 3 times that of Brazil, India, or South Africa.

Figure 1 shows that Musculoskeletal & Pain conditions contributed the most to physical rehabilitation needs in 2017 across the five countries, ranging from 36% of South African’s physical rehabilitation needs to 60% of Brazilian’s - greater than any global benchmark. In India and Brazil, cardiothoracic conditions were the second most represented (22% and 16%, respectively). In China, both neurological and cardiothoracic conditions hold that second rank (18% each). In Russia, neurological conditions were the second most represented (19%). Finally, in South Africa, HIV-related conditions were the second most representative (25%): in no other BRICS country did HIV-related conditions accounted for more than 1% of physical rehabilitation needs, and the maximum global benchmark was 6% for low-income countries. 

Finally, Table 4 shows how the distribution of physical rehabilitation needs evolved per condition groups from 1990 to 2017. In South Africa, physical rehabilitation needs coming from HIV-related conditions increased massively, yet logarithmically: from 1% to 25% of South African’s physical rehabilitation needs (*p* < 0.01). In China, the percentage of physical rehabilitation needs coming from both neurological conditions and neoplasms increased significantly (from 12.3% to 18.4%, and from 1.42% to 2.86%, respectively: *p* < 0.01), both with an exponential type of growth. In India, the percentage of physical rehabilitation needs coming from pediatric conditions increased significantly (from 15.7% to 17.8%: *p* < 0.01). In Russia, the percentage of physical rehabilitation needs that came from neoplasms has grown by 47% (*p* < 0.01). Finally, only in Brazil did the percentage of physical rehabilitation needs coming from musculoskeletal & pain conditions increased significantly (from 57.5% to 59.9%; *p* < 0.01), becoming greater than that in high-income countries (57.7% in 2017).

## 4. Discussion

In each of the BRICS countries, total physical rehabilitation needs have increased significantly from 1990 to 2017 in absolute values, per-capita, and in percentage of all YLDs. This means that, in each of these countries, physical rehabilitation needs have increased beyond the population growth, and that physical rehabilitation could be helpful for a greater portion of non-fatal health losses. 

Apart from common trends across the BRICS nations (e.g., growth of total physical rehabilitation needs per capita with no decrease or even an increase in age-standardized needs), we found important country-specific differences across the BRICS countries in the amount, typology, and evolution of their physical rehabilitation needs.

For the overall age-standardized YLD Rates germane to physical rehabilitation, we did not observe significant changes for India and Russia. This means that for these countries the aging of the population (and the subsequent higher disability rates [23,26,28]) has been a key driver of their increased physical rehabilitation needs, including in YLDs Rates. Yet, in Brazil, China, and South Africa we did observe a significant growth in the age-standardized YLD Rates, which means that variables other than those related to the population ageing might have contributed to the overall growth of their physical rehabilitation needs. Only in high-income nations did we observe a significant decrease in the rehabilitation-related age-standardized YLD Rates. Possibly a more developed rehabilitation infrastructure or health care systems in high-income nations than in the BRICS nations contributed their reduction in age-standardized YLD-rates for rehabilitation-related conditions.

China stood out with the greatest amount and an exponential type of growth in the physical rehabilitation needs from neurological and neoplasm conditions. The population ageing, derived from the previous one-child policy, increased life expectancy [30], increasing survival rates for those with neoplasm or other health conditions, along with the huge baby boom cohorts born in 1950s and 1960s entering old ages [39,40,51], can partly account for these findings. As survival rates from health conditions likely will increase further [52] and life expectancy in China is projected to surpass 80 years by 2040 [53], the rise of physical rehabilitation needs in China being observed is likely to continue into the future. Moreover, the meeting the rehabilitation needs of older adults in the rural, underserved regions of China can be particularly challenging as rehabilitation services typically are distant and/or scant and family support is increasingly absent (e.g., much of the working-age population has moved to urban, industrialized areas) [24]. Caregiver-delivered, digital-supported, and nurse-led interventions have been trialed to close the rehabilitation service gap in rural China, but more work is needed to achieve optimal results [54]. These will be important needs and gaps for China to address through future research and policy development [24,51,54,55].

In India, another highly populated and emerging economy in Asia, the typology of physical rehabilitation needs was different than for the other BRICS nations and, in some respects, closer to that of lower income countries. One such example is the absolute and relative growth of physical rehabilitation needs arising from pediatric conditions. This pattern may reflect different economic status from the other BRICS: i.e., although an emergent economy, India is still a lower-middle income nation per the World Bank classification while the BRICS counterparts are UMICs. Alternatively, the findings may reflect a different population ageing structure and higher fertility rates [31]. In addition to the particular rise of pediatric physical rehabilitation needs in India, YLD Rates increased for each other major groups of conditions responsive to physical rehabilitation. Indeed, the epidemiological transition for higher rates of non-communicable, chronic and disabling conditions has been impacting India, although differentially across regions [13,56,57,58]. All these needs contrast with the existing systems for rehabilitation and social care in the country. There is an acute shortage of 6.4 million allied health professionals in India [59]. There are no professional bodies that regulate the practice or practice standards for any health professionals [60]. The national program for tracking of non-communicable health conditions focuses on early detection and treatment [61]. Furthermore, the health system’s infrastructure is not architecturally and socially accessible to people with disabilities [62]. Overall, the health system has not been capable of meeting the growing need for physical rehabilitation in India [58,63]. Similar to China, technologically-enabled service delivery solutions have been trialed to meet the growing physical rehabilitation needs in India [64]. Interventions like these needs to be tested for scalability in combination with existing health and rehabilitation services in India.

South Africa, a leading UMIC within the African continent, more than doubled their absolute physical rehabilitation needs, mimicking the trend in low-income countries. This increase in need for rehabilitation is partly driven by the HIV/AIDS endemic. HIV-related conditions accounted for one-quarter of South African’s physical rehabilitation needs in 2017, compared to 1% in any of the four other BRICS countries. South Africa’s successful roll-out of highly active antiretroviral therapy (HAART) has transformed HIV into a chronic disease, and people with HIV can now achieve normal life-expectancy [65]. An increasing number of South Africans with HIV live longer, but with either the potential for or already established impairments in body structure such as muscle weakness, and that may cause limitations in activities of daily living and restrictions in participation [65]. Despite this, South Africa’s HIV policies and guidelines do not speak to HIV-related disabilities, as premature mortality remains a key national health indicator [66]. However, political will to address non-fatal health loss is rising [67] as an increasing body of literature signals the need to address HIV-related disability [68,69,70]. Increased investment in health resources to enhance the quality of life and functioning in people with HIV will require concerted effort. Access to adequate rehabilitation has the potential to optimize functioning, employability and could even enhance HAART adherence in people living with HIV [70].

Russia had the highest rate of physical rehabilitation needs in 2017 (6393 YLDs per 100,000 inhabitants), but the lowest percent change since 1990; a pattern closest to that of high-income countries more than other BRICS nations. Similarly, we found that Russia had the highest rates, although not the highest growth, in physical rehabilitation needs from neurological conditions, including about the triple of those from Brazil, India and South Africa. The persistently high burden of rehabilitation-sensitive conditions in Russia may be a result of several factors. First, the incidence of major chronic non-communicable diseases remains high; national statistics have shown, for example, that the incidence of coronary heart disease has increased substantially from 495 to 701 per 100,000 inhabitants from 2010 to 2016 [71]. That accounts for the high prevalence of risk factors, primarily of hypertension and overweight/obesity, which is on the rise, especially in men [72]. In contrast, the rates of smoking and harmful alcohol use are currently decreasing [73], although historically high [14]. Secondly, the overall quality of healthcare has increased in Russia, resulting in increased survival; for example, the age-standardized mortality rates from myocardial infarction decreased from 47.1 to 42.9 per 100,000 inhabitants [2012–2016] [74]. Thirdly, higher survival rates may also arise from screening and early diagnosis programs such as the national universal health screening program for cancers and government-led program for screening cardiovascular risk factors and diseases [75]. Fourthly, local traditions of ICD 10 codes interpretation may lead to inappropriate coding of some dementia cases as cardiovascular or neurologic conditions instead of mental disorders [76]. Finally, the population has aged in Russia, which is not surprising given the growing per capita income in recent years, and the population aging seen in high-income countries [77]. Relatedly, the westernization of lifestyle in Russia, with a greater availability of highly processed foods and environmental problems due to increased car traffic, likely is playing in rendering physical rehabilitation needs in Russia similar to those high-income nations. To help meet their nation’s high physical rehabilitation needs, Russia has been actively developing their medical rehabilitation paradigm [78] and infrastructure [79,80].

Finally, for Brazil, we found that conditions responsive to physical rehabilitation currently account for about two-thirds of the nation’s YLD (no other BRICS nation came close to 50%). This means that physical rehabilitation can address a larger portion of the country’s non-fatal health losses when compared to the BRICS counterparts. Brazil also stood out with the highest portion of physical rehabilitation needs coming from musculoskeletal & pain conditions (60%), and with substantial growth in this percent value over time. Key explanations for that finding may include the prevalence of interpersonal violence in Brazil [16], and the high and rising prevalence of road traffic injuries, especially associated with high consumption of alcohol involving young pedestrian and motorcyclists within urbanized environments [81,82]. This study emphasizes not only the need of expanding Brazilian public policies to ameliorate external causes of injury as well as chronic disease prevention, but also for implementing a rehabilitation infrastructure capable of addressing the growing burden of physical impairments in Brazil. 

### Study Limitations

This study has the following limitations:First, YLDs from selected health conditions are but proxy indicators of physical rehabilitation needs, i.e., not a direct functional or impairments-based measure. Nonetheless, YLDs is the aggregative measure of non-fatal health loss from the prominent GBD study and includes variables such as the prevalence of conditions, the time lived with sequalae from the respective conditions, and weighted for the appraised severity of those sequelae.Second, the set of conditions whose YLDs likely benefit from rehabilitation were replicated from a previous study which systematically reviewed evidence linking those conditions to rehabilitation needs; [25] nonetheless, these conditions cannot be considered a fixed standard as the relevant conditions may change over time with the advancement of rehabilitation therapies and their scientific support. For example, the recent COVID-19 pandemic has been boosting new types of rehabilitation need (e.g., for respiratory therapy; for the rehabilitation for the post-intensive care syndrome) [83,84,85], which were not reflected in the data up to 2017.Third, YLD values (extracted from the GBD 2017) are only estimates based on the best-available evidence, not actual YLDs. The GBD 2017 is the most comprehensive epidemiological study to date, and the amount of data used to create those estimates is unprecedented [25,48]. Even so, the quality and the quantity of the underlying data for computing the GBD estimates vary across locations and in time within the same location, which in turn affects the precision of the YLD estimates. However, lower precision does not equate to bias toward over or under-estimation of YLDs for the earlier times or for the locations in which less or lower-quality data were available. At each cycle, the GBD study (e.g., the GBD 2017) apply the new data and more advanced estimation methods to re-calculate YLDs across locations and the entire time series (since 1990), not only the values for 2017.Fourth, most data obtained for the GBD study (e.g., in India) are from self-reports and hence many undiagnosed conditions might not be included within this data to represent the true picture. Hence results of this study could be a gross under-estimation of the problem, at least in the absolute values.Fifth, we did not extract or analyze sub-national data (e.g., Brazilian states), although data are available for that from the GBD 2017 and some important differences exist in both economic and epidemiological profiles across regions or states of the analyzed countries [13,16,39].Sixth, we do not supplement our analyses of physical rehabilitation ‘needs’ with indicators of physical rehabilitation ‘supply’ across nations, the other key element in the resources planning equation. In part, this follows the lack of available data. For example, the World Confederation of Physical Therapy reports data on the amount of practicing physical therapists per nations, as locally collected from authoritative sources or estimated by national associations (i.e., their member organizations) for a total of 89 countries, but unfortunately not from 3 of the analyzed countries (China, Russia, or India) [86].

## 5. Conclusions

Physical rehabilitation needs have increased significantly from 1990 to 2017 in each of the BRICS nations, both in absolute and relative values. However, apart from the common trend in overall growth, each of the BRICS nations had own patterns for the amount, typology, and evolution of their physical rehabilitation needs. The BRICS nations and coalition need to address the common challenge of planning for and deploying the required resources for meeting the growing physical rehabilitation needs of their population, at the same time they look at country-specific challenges such as the physical rehabilitation needs coming from HIV/AIDs-related conditions in South Africa, pediatric conditions in India, musculoskeletal conditions in Brazil, and neurological conditions in Russia and China. This study shows that physical rehabilitation needs can be determined and compared across nations, and hence can be used to inform rehabilitation resources and service planning. Most importantly, this study makes clear that physical rehabilitation needs and growth patterns may not be assumed equal across nations, irrespective of similarities in income or emerging development.

## Figures and Tables

**Figure 1 ijerph-17-04139-f001:**
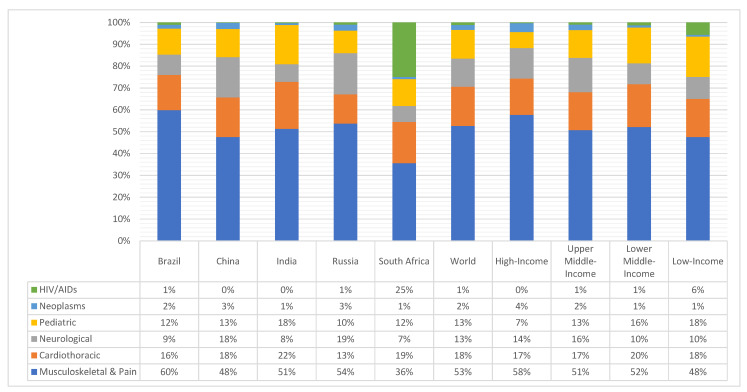
How YLD Rates (per 100,000 people) likely benefiting from physical rehabilitation are distributed per major groups of conditions in 2017 across the five nations analyzed, and how these values compare to global benchmarks.

**Table 1 ijerph-17-04139-t001:** How Years Lived with Disability from health conditions of the Global Burden of Disease Study 2017 are grouped, referring to type of impairments or type physical rehabilitation service level. Due to its specificity, we do not aggregated YLDs from neoplasm and HIV/AIDS.

Underlying Health Conditions (from the Global Burden of Disease Study)	Intermediate Aggregation	Main Groups (Condition Types)
Low back pain	Pain	Musculoskeletal & Pain
Neck pain		
Tension-type headaches		
Injuries (all selected except Spinal & Brain Injuries; Asphyxiation; and Severe Chest Injuries)	Musculoskeletal Trauma	
Osteoarthritis	Musculoskeletal disorders	
Rheumatoid arthritis		
Gout & Other musculoskeletal disorders		
Leprosy		
Chronic Respiratory Diseases	Pulmonary	
Severe Chest Injuries		
Cardiovascular Diseases (excluding Stroke)	Cardiovascular	Cardiotoracic
Heart Failure (resulting from all the non-considered health conditions)		
Stroke	Neurological Disorders (non-communicable)	Neurological
Multiple sclerosis; Parkinson’s; Alzheimer’s & Other Dementias; Motor neuron disease; Other neurological disorders; Neoplasm–brain & nervous system		
Infectious–affecting the nervous system: Encephalitis; Meningitis; Tetanus; ZIKA virus	Neurological–Infectious *	
Syndrome: Guillain-Barré (resulting from non-considered health conditions)		
Spinal Cord Injury	Neurological Trauma	
Traumatic Brain Injury; Asphyxiation		
Congenital birth defects (*digestive & urogenital disorders excluded*)	-	Pediatric **
Neonatal		
Autism Spectrum Disorder		
All Neoplasms (not nervous system)	-	Neoplasm (not nervous system)
HIV/AIDS	-	HIV-related

Legend: * Conditions that may have early onsets and hence rather require *pediatric* physical rehabilitation. Zika virus can led to both neurological and musculoskeletal sequalae and physical rehabilitation interventions. ** When leading to long-term impairments, *pediatric* conditions may also require adult physical rehabilitation services.

**Table 2 ijerph-17-04139-t002:** YLDs (Years Lived with Disability), in four metrics, from all conditions likely benefiting from physical rehabilitation–i.e., all conditions combined. YLD values are provided for each of the five countries analyzed, as well as for global benchmarks.

	1990	2017	% Change [1990–2017]	Regression Model Type	*r* ^2^	*b* Coefficient	95% CI	99% CI
**YLD Counts, Millions**								
Brazil	4.94	9.02	82.6%	Linear	1	0.16 *	0.15–0.16	0.15–0.16
China	39.8	67.0	68.1%	Linear	0.98	1.01 *	0.94–1.07	0.92–1.10
India	28.8	55.1	91.6%	Linear	0.98	0.98 *	0.93–1.03	0.91–1.05
Russia	8.69	9.35	7.6%	Linear	0.76	0.03 *	0.02–0.04	0.02–0.04
South Africa	1.25	2.68	114.1%	Linear	0.94	0.06 *	0.06–0.07	0.05–0.07
*World*	*206.4*	*342.9*	*66.2%*	*Linear*	*0.99*	*5.10 **	*4.88–5.32*	*4.80–5.40*
*High-income*	*57.5*	*79.0*	*37.4%*	*Linear*	*0.99*	*0.81 **	*0.77–0.84*	*0.76–0.86*
*Upper Middle-Income*	*75.9*	*123.0*	*62.1%*	*Linear*	*0.99*	*1.78 **	*1.69–1.87*	*1.66–1.90*
*Lower Middle-Income*	*62.3*	*118.8*	*90.4%*	*Linear*	*0.99*	*2.10 **	*2.02–2.19*	*1.99–2.22*
*Low-Income*	*9.81*	*20.8*	*111.5%*	*Linear*	*1*	*0.39 **	*0.38–0.40*	*0.38–0.41*
**YLD Rates (per 100,000 inhabitants)**								
Brazil	3306	4528	28.8%	Linear	1	37.2 *	36.3–38.1	36.0–38.4
China	3329	4743	42.5%	Linear	0.95	54.3 *	49.3–59.2	47.6–60.9
India	3300	3990	20.9%	Linear	0.92	25.2 *	22.2–28.2	21.1–29.3
Russia	5741	6393	11.4%	Linear	0.91	31.6 *	27.6–35.6	26.2–37.0
South Africa	3399	4803	43.3%	Linear	0.85	73.4 *	61.1–85.7	56.8–90.0
World	*3825*	*4488*	*17.3%*	*Linear*	*0.96*	*25.7 **	*23.7–27.7*	*23.0–28.4*
*High-income*	*5748*	*6643*	*15.6%*	*Linear*	*0.98*	*33.1 **	*31.1–35.1*	*30.4–35.8*
*Upper Middle-Income*	*3594*	*4669*	*29.9%*	*Linear*	*0.96*	*42.6 **	*39.3–46.0*	*38.1–47.2*
*Lower Middle-Income*	*3233*	*3806*	*17.7%*	*Linear*	*0.96*	*21.6 **	*19.9–23.2*	*19.3–23.8*
*Low-Income*	*2977*	*3112*	*4.5%*	*Logarithmic*	*0.50*	*2.5 ***	*0.40–4.55*	*−0.33–5.28*
**Age-standardized YLD Rates**								
Brazil	3993	4010	0.44%	Linear	0.69	2.9 *	2.08–3.63	1.81–3.90
China	3795	3898	2.71%	Linear	0.24	5.4 *	1.47–9.02	0.14–10.4
India	4361	4368	0.16%	Logarithmic	0.09	- 0.2	−2.62–2.22	−3.47–3.08
Russia	5156	4991	−3.20%	Logarithmic	0.13	- 0.1	−4.02–3.77	−5.40–5.14
South Africa	4415	5131	16.2%	Logarithmic	0.72	47.8 *	32.7–62.8	27.5–68.1
*World*	*4377*	*4334*	*−1.0%*	*Logarithmic*	*0.22*	*−0.62*	*−2.13–0.89*	*−2.66–1.42*
*High-income*	*5007*	*4872*	*−2.7%*	*Logarithmic*	*0.86*	*−5.36 **	*−6.76–(−3.96)*	*−7.26–(−3.47)*
*Upper Middle-Income*	*4106*	*4080*	*−0.6%*	*Linear*	*0.04*	*1.34*	*−1.38–4.06*	*−2.33–5.02*
*Lower Middle-Income*	*4262*	*4314*	*1.2%*	*Linear*	*0.46*	*2.33 **	*1.26–3.40*	*0.89–3.78*
*Low-Income*	*4189*	*4276*	*2.1%*	*Logarithmic*	*0.15*	*0.29*	*−3.29–3.87*	*−4.55–5.14*
**% of YLDs Benefiting from Physical Rehabilitation (among total YLDs)**								
Brazil	60.1%	66.2%	10.2%	Linear	0.96	0.26 *	0.24–0.28	0.23–0.29
China	37.8%	44.8%	18.7%	Linear	0.97	0.26 *	0.24–0.28	0.24–0.29
India	29.9%	35.9%	20.1%	Linear	0.93	0.23 *	0.21–0.26	0.20–0.27
Russia	45.4%	46.6%	2.6%	Linear	0.46	0.07 *	0.04–0.09	0.03–0.11
South Africa	34.8%	43.8%	25.8%	Linear	0.87	0.43 *	0.37–0.50	0.34–0.52
*World*	*36.7%*	*40.2%*	*9.5%*	*Linear*	*0.97*	*0.14 **	*0.13–0.15*	*0.12–0.15*
*High-income*	*47.6%*	*48.6%*	*2.2%*	*Linear*	*0.87*	*0.03 **	*0.03–0.04*	*0.02–0.04*
*Upper Middle-Income*	*37.9%*	*42.2%*	*11.4%*	*Linear*	*.97*	*0.17 **	*0.16–0.18*	*0.15–0.19*
*Lower Middle-Income*	*30.7%*	*35.9%*	*16.8%*	*Linear*	*0.98*	*0.20 **	*0.19–0.21*	*0.19–0.22*
*Low-Income*	*27.8%*	*32.1%*	*15.4%*	*Linear*	*0.97*	*0.16 **	*0.15–0.17*	*0.14–0.17*

Data obtained from: http://ghdx.healthdata.org/gbd-results-tool. Abbreviations: YLD–Year Lived with Disability. Legend: * *p* < 0.01; ** *p* < 0.05. While possibly obtained through the same source, data for the global benchmarks, including computed values, were extracted from: Jesus TS, Landry MD, Hoenig H. Global need for physical rehabilitation: systematic analysis from the Global Burden of Disease Study 2017. *Int J Environ Res Public Health*. 2019, 16: 980; Notes: The “*b* coefficient” refers to the annual change within a linear regression model. Different population structures apply to countries with varying income levels; so, cross-location comparisons are not valid for the metric YLD Counts, except for the variable “% change [1990–2017”.

**Table 3 ijerph-17-04139-t003:** How YLD Rates (i.e., Years Lived with Disability per 100,000 people) likely benefiting from physical rehabilitation are distributed per major groups of conditions across the five countries analyzed, and how values have evolved over time [1990–2017].

	1990	2017	% Change [1990–2017]	Regression Model Type	r^2^	b Coefficient	95% CI	99% CI
**Musculoskeletal & Pain**								
Brazil	1901	2551	34.2%	Linear	1	25.4 *	24.9–25.9	24.8–26.0
China	1646	2258	37.2%	Linear	0.99	25.0 *	23.8–26.2	23.4–26.6
India	1765	2048	16.0%	Linear	0.91	10.1 *	8.9–11.4	8.4–11.9
Russia	3271	3435	5.0%	Linear	0.88	11.6 *	9.8–13.3	9.2–13.9
South Africa	1605	1731	7.9%	Linear	0.93	5.7 *	5.1–6.4	4.9–6.6
*World*	*2071*	*2363*	*14.1%*	*Linear*	*0.98*	*11.4 **	*10.7−12.1*	*10.5−12.4*
*High-Income*	*3359*	*3835*	*14.2%*	*Linear*	*0.99*	*16.8 **	*16.1−17.5*	*15.9−17.7*
*Upper Middle-Income*	*1875*	*2369*	*26.3%*	*Linear*	*0.98*	*20.5 **	*19.4−21.5*	*19.0−21.9*
*Lower Middle-Income*	*1724*	*1983*	*15.1%*	*Linear*	*0.96*	*9.4 **	*8.6−10.3*	*8.3−10.5*
*Low-Income*	*1486*	*1491*	*0.4%*	*Linear*	*0.09*	*−0.6*	*−1.3−0.1*	*−1.5−0.4*
**Neurological**								
Brazil	263	396	50.4%	Linear	0.96	5.2 *	4.7–5.6	4.6–5.7
China	410	870	112.5%	Exponential	0.97	15.6 *	13.8–17.3	13.2–17.9
India	247	323	30.5%	Linear	0.94	2.8 *	2.5–3.1	2.4–3.2
Russia	922	1207	30.9%	Linear	0.83	11.4 *	9.3–13.4	8.6–14.1
South Africa	330	356	7.9%	Linear	0.57	0.7 *	0.5–1.0	0.4–1.1
*World*	*441*	*578*	*31.1%*	*Linear*	*0.89*	*4.8 **	*4.1−5.5*	*3.9−5.7*
*High-Income*	*750*	*929*	*23.7%*	*Linear*	*0.90*	*6.1 **	*5.3−6.9*	*5.0−7.2*
*Upper Middle-Income*	*441*	*736*	*66.9%*	*Exponential*	*0.95*	*10.4 **	*9.1−11.6*	*8.7−12.0*
*Lower Middle-Income*	*301*	*364*	*21.2%*	*Linear*	*0.86*	*2.3 **	*1.9−2.7*	*1.8−2.8*
*Low-Income*	*321*	*314*	*−2.1%*	*Linear*	*0.53*	*−0.5 **	*−0.7−(−0.3)*	*−0.8−(−0.2)*
**Cardiotoracic**								
Brazil	624	685	9.8%	Linear	0.88	2.65 *	2.3–3.1	2.11–3.19
China	759	857	12.9%	Linear	0.36	4.95 *	2.3–7.6	1.35–8.56
India	746	857	14.8%	Linear	0.49	4.02 *	2.4–5.7	1.78–6.26
Russia	802	856	6.7%	Linear	0.94	2.21 *	2.0–2.4	1.90–2.52
South Africa	851	918	7.9%	Logarithmic	0.73	2.89 *	1.7–4.1	1.23–4.56
*World*	*733*	*807*	*10.1%*	*Linear*	*0.63*	*3.5 **	*2.4−4.5*	*2.0−4.9*
*High-Income*	*956*	*1103*	*15.4%*	*Linear*	*0.94*	*7.0 **	*6.3−7.7*	*6.0−8.0*
*Upper Middle-Income*	*719*	*810*	*12.7%*	*Linear*	*0.55*	*4.3 **	*2.7−5.9*	*2.2−6.4*
*Lower Middle-Income*	*662*	*745*	*12.5%*	*Linear*	*0.67*	*3.4 **	*2.5−4.4*	*2.1−4.7*
*Low-Income*	*557*	*549*	*−1.5%*	*Linear*	*0.06*	*−0.3*	*−0.8−0.2*	*−1.0−0.3*
**Pediatric**								
Brazil	468	505	7.8%	Linear	0.99	1.48 *	1.42–1.54	1.40–1.57
China	467	613	31.4%	Linear	0.97	5.46 *	5.04–5.87	4.90–6.02
India	519	713	37.3%	Linear	0.99	7.18 *	6.92–7.43	6.83–7.52
Russia	636	658	3.6%	Linear	0.50	1.89 *	1.13–2.66	0.86–2.93
South Africa	558	596	6.8%	Logarithmic	0.06	−0.03	−1.19–1.12	−1.60–1.53
*World*	498	588	18.0%	Linear	0.99	3.5 *	3.4−3.6	3.3−3.7
*High-Income*	493	486	−1.5%	Logarithmic	0.91	−0.2 *	−0.3−(−0.2)	−0.3−(−0.1)
*Upper Middle-Income*	505	594	17.7%	Linear	0.97	3.4 *	3.2−3.6	3.1−3.7
*Lower Middle-Income*	507	624	23.0%	Linear	0.99	4.6 *	4.4−4.8	4.4−4.8
*Low-Income*	424	579	36.5%	Linear	0.97	6.4 *	5.9−6.8	5.8−6.9
**Neoplasm**								
Brazil	38	75	99%	Linear	1	1.32 *	1.30–1.35	1.29–1.36
China	47	136	189%	Exponential	0.94	3.00 *	2.51–3.49	2.34–3.66
India	22	34	58%	Linear	0.81	0.40 *	0.32–0.47	0.29–0.05
Russia	106	174	64%	Linear	0.93	2.68 *	2.38–2.98	2.27–3.08
South Africa	38	49	30%	Logarithmic	0.39	0.28 **	0.03–0.53	−0.06–0.62
*World*	*62*	*100*	*62.3%*	*Linear*	*0.95*	*1.3 **	*1.1−1.4*	*1.1−1.4*
*High-Income*	*173*	*271*	*56.6%*	*Linear*	*0.99*	*3.4 **	*3.3−3.5*	*3.2−3.6*
*Upper Middle-Income*	*50*	*115*	*130%*	*Exponential*	*0.95*	*2.3 **	*1.9−2.6*	*1.8−2.7*
*Lower Middle-Income*	*27*	*38*	*40.1%*	*Linear*	*0.76*	*0.4 **	*0.3−0.5*	*0.3−0.5*
*Low-Income*	*27*	*26*	*−1.9%*	*Linear*	*0.55*	*−0.1 **	*−0.2−(−0.1)*	*−0.2−(−0.1)*
**HIV-related**								
Brazil	12.4	45.2	163%	Linear	0.94	1.1 *	1.0–1.2	0.09–1.3
China	1.2	7.4	528%	Linear	0.98	0.2 *	0.2–0.2	0.2–0.3
India	0.8	15.2	1905%	Logarithmic	0.64	0.7 *	0.3–1.0	0.2–1.1
Russia	4.5	61.7	1277%	Exponential	1	1.8 *	1.5–2.1	1.4–2.2
South Africa	17.3	1219	6469%	Logarithmic	0.83	63.8 *	51.0–76.5	46.5–81.0
*World*	*17*	*54*	*207%*	*Logarithmic*	*0.79*	*1.28 **	*0.79−1.77*	*0.62−1.94*
*High-Income*	*14*	*19*	*37%*	*Linear*	*0.04*	*0.04 **	*−0.2−0.11*	*−0.6−0.13*
*Upper Middle-Income*	*4*	*48*	*1255%*	*Linear*	*0.89*	*1.89 **	*1.62−2.15*	*1.53−2.24*
*Lower Middle-Income*	*23*	*50*	*113%*	*Logarithmic*	*0.71*	*1.39 **	*0.85−1.93*	*0.66−2.12*
*Low-Income*	*160*	*178*	*12%*	*Linear*	*0.09*	*−2.29*	*−5.15−0.58*	*−6.16−1.59*

Data obtained from: http://ghdx.healthdata.org/gbd-results-tool. Abbreviations: YLD–Year Lived with Disability. Legend: * *p* < 0.01; ** *p* < 0.05. While possibly obtained through the same source, data for the global benchmarks, including computed values, were extracted from: Jesus TS, Landry MD, Brooks D, Hoenig H. Physical rehabilitation needs per condition type: Results from the Global Burden of Disease study 2017. Arch Phys Med Rehabill. doi: 10.1016/j.apmr.2019.12.020; Notes: The “*b* coefficient” refers to the annual change within a linear regression model. HIV/AIDs include YLDs from resultant tuberculosis.

**Table 4 ijerph-17-04139-t004:** How YLD Rates (Years Lived with Disability per 100,000 people) likely benefiting from physical rehabilitation are distributed per major groups of conditions across the five countries analyzed, and how those percent values have evolved over time [1990–2017].

	# 1990	# 2017	% Change [1990–2017]	Regression Model Type	r^2^	b Coefficient	95% CI	99% CI
**Musculoskeletal & Pain**								
Brazil	57.5%	59.9%	4.2%	Linear	0.97	0.09 *	0.09–0.10	0.09–0.10
China	49.4%	47.6%	−3.7%	Linear	0.09	−0.03	−0.08–(−0.01)	−0.08–0.01
India	53.5%	51.3%	−4.0%	Linear	0.91	−0.09	−0.10–(−0.08)	−0.10–(−0.07)
Russia	57.0%	53.7%	−5.7%	Linear	0.67	−0.10	−0.13–(−0.07)	−0.14–(−0.06)
South Africa	47.2%	35.6%	−24.7%	Logarithmic	0.87	−0.54 *	−0.66–(−0.41)	−0.70–(−0.37)
World	54.1%	52.6%	−2.8%	Linear	0.87	−0.06 *	−0.06−(−0.05)	−0.07−(−0.03)
High–Income	58.4%	57.7%	−1.2%	Linear	0.72	−0.04 *	−0.05−(−0.03)	−0.05−(−0.03)
Upper Middle–Income	52.2%	50.7%	−2.8%	Linear	0.29	−0.04 *	−0.06−(−0.01)	−0.07−(−0.01)
Lower Middle–Income	53.3%	52.1%	−2.3%	Linear	0.97	−0.06 *	−0.06−(−0.05)	−0.06−(−0.05)
Low–Income	49.9%	41.7%	−4.0%	Logarithmic	0.62	−0.06 *	−0.09−(−0.02)	−0.11−(−0.01)
**Neurological**								
Brazil	8.0%	9.3%	16.8%	Linear	0.86	0.05 *	0.04–0.06	0.04–0.06
China	12.3%	18.4%	49.2%	Exponential	0.96	0.19 *	0.17–0.21	0.16–0.22
India	7.5%	8.1%	8.0%	Linear	0.95	0.02 *	0.02–0.03	0.02–0.03
Russia	16.1%	18.9%	17.6%	Linear	0.74	0.10 *	0.08–0.12	0.07–0.13
South Africa	9.7%	7.3%	−24.7%	Logarithmic	0.81	−0.12 *	−0.15–(−0.09)	−0.16–(−0.08)
World	11.5%	12.9%	11.7%	Linear	0.72	0.04 *	0.03−0.05	0.03−0.05
High–Income	13.1%	14.0%	7.1%	Linear	0.58	0.03 *	0.02−0.04	0.02−0.04
Upper Middle–Income	12.3%	15.8%	28.4%	Exponential	0.90	0.11 *	0.10−0.13	0.10−0.13
Lower Middle–Income	9.3%	9.6%	2.9%	Linear	0.23	0.01 *	0.002–0.02	0.002–0.02
Low–Income	10.8%	10.1%	−6.3%	Logarithmic	0.86	0.02 *	−0.03−(−0.02)	−0.03−(−0.02)
**Cardiothoracic**								
Brazil	18.9%	16.1%	−14.7%	Linear	0.96	−0.10 *	−0.11–(−0.09)	−0.11–(−0.09)
China	22.8%	18.1%	−20.8%	Logarithmic	0.71	−0.16 *	−0.21–(−0.11)	−0.23–(−0.09)
India	22.6%	21.5%	−5.0%	Logarithmic	0.46	−0.04 **	−0.07–(−0.01)	−0.07–0.002
Russia	14.0%	13.4%	−4.2%	Linear	0.79	−0.04 *	−0.04–(−0.03)	−0.05–(−0.03)
South Africa	25.0%	18.9%	−24.7%	Linear	0.89	−0.30 *	−0.34–(−0.26)	−0.36–(−0.24)
World	19.2%	18.0%	−6.2%	Logarithmic	0.57	−0.03 *	−0.05−(−0.01)	−0.06−(−0.01)
High–Income	16.6%	16.6%	−0.1%	Linear	0.56	0.03 *	0.02−0.03	0.01−0.04
Upper Middle–Income	20.0%	17.4%	−13.2%	Logarithmic	0.74	−0.09 *	−0.12−(−0.06)	−0.13−(−0.05)
Lower Middle–Income	20.5%	19.6%	−4.4%	Logarithmic	0.48	−0.02 **	−0.04−(−0.005)	−0.05−0.002
Low–Income	18.7%	17.6%	−5.7%	Logarithmic	0.43	−0.02	−0.05−0.001	−0.06−0.01
**Pediatric**								
Brazil	14.2%	11.9%	−16.3%	Linear	1	−0.09 *	−0.09–(−0.09)	−0.09–(−0.09)
China	14.0%	12.9%	−7.8%	Linear	0.39	−0.05 *	−0.08–(−0.03)	−0.09–(−0.02)
India	15.7%	17.9%	13.6%	Linear	0.84	0.08 *	0.07–0.10	0.06–0.10
Russia	11.1%	10.3%	−7.0%	Logarithmic	0.92	−0.02 *	−0.09–(−0.09)	−0.09–(−0.09)
South Africa	16.4%	12.3%	−25.5%	Logarithmic	0.80	−0.23 *	−0.29–(−0.16)	−0.31–(−0.14)
World	13.0%	13.1%	0.6%	Logarithmic	0.15	0.003	−0.004−0.011	−0.007−0.013
High–Income	8.6%	7.3%	−14.8%	Linear	0.99	−0.05 *	−0.048−(−0.045)	−0.048−(−0.044)
Upper Middle–Income	14.0%	12.7%	−9.4%	Linear	0.73	−0.06 *	−0.073−(−0.044)	−0.078−(−0.039)
Lower Middle–Income	15.7%	16.4%	4.5%	Logarithmic	0.74	0.03 *	0.023−0.042	0.020−0.045
Low–Income	14.2%	18.6%	30.6%	Linear	0.92	0.19 *	0.171−0.216	0.163−0.224
**Neoplasms**								
Brazil	1.14%	1.75%	54%	Linear	0.98	0.021 *	0.020–0.022	0.019–0.022
China	1.41%	2.86%	103%	Exponential	0.91	0.048 *	0.040–0.056	0.037–0.059
India	0.66%	0.86%	30%	Linear	0.71	0.006 *	0.004–0.007	0.004–0.008
Russia	1.84%	2.72%	47%	Linear	0.87	0.032 *	0.027–0.032	0.025–0.039
South Africa	1.10%	1.00%	−10%	Linear	0.56	−0.011 *	−0.015–(−0.007)	−0.016–(−0.006)
World	1.6%	2.2%	38.3%	Linear	0.94	0.02 *	0.02−0.02	0.02−0.02
High–Income	3.0%	4.1%	35.5%	Logarithmic	0.97	0.04 *	0.03−0.04	0.03−0.04
Upper Middle–Income	1.4%	2.5%	77.0%	Exponential	0.92	0.04 *	0.03−0.04	0.03−0.04
Lower Middle–Income	0.8%	1.0%	19.0%	Linear	0.53	0.01 *	0.004−0.01	0.003−0.01
Low–Income	0.9%	0.8%	−6.1%	Linear	0.59	−0.01 *	−0.01−(−0.004)	−0.01−(−0.003)
**HIV–related**								
Brazil	0.38%	1.06%	182%	Linear	0.89	0.022 *	0.019–0.025	0.018–0.027
China	0.04%	0.16%	341%	Linear	0.99	0.005 *	0.005–0.005	0.004–0.005
India	0.02%	0.38%	1558%	Linear	0.55	0.016 *	0.006–0.026	0.003–0.030
Russia	0.08%	0.97%	1137%	Exponential	1	0.028 *	0.024–0.033	0.023–0.034
South Africa	0.51%	25.0%	4831%	Logarithmic	0.85	1.193 *	0.939–1.448	0.849–1.538
World	0.5%	1.2%	162%	Linear	0.89	0.02 *	0.012−0.037	0.007−0.042
High–Income	0.2%	0.3%	19%	Linear	0.99	0.001	−0.002−0.001	−0.003−0.001
Upper Middle–Income	0.1%	1.0%	943%	Logarithmic	0.55	0.04 *	0.032−0.048	0.029−0.050
Lower Middle–Income	0.7%	1.3%	81%	Exponential	1.0	0.03 *	0.016−0.050	0.010−0.056
Low–Income	5.4%	5.7%	7%	Logarithmic	0.85	−0.08	−0.166−0.009	−0.197−0.040

Data obtained from: http://ghdx.healthdata.org/gbd-results-tool. Abbreviations: YLD–Year Lived with Disability. Legend: * *p* < 0.01; ** *p* < 0.05. While possibly obtained through the same source, data for the global benchmarks, including computed values, were extracted from: Jesus TS, Landry MD, Brooks D, Hoenig H. Physical rehabilitation needs per condition type: Results from the Global Burden of Disease study 2017. Arch Phys Med Rehabill. doi: 10.1016/j.apmr.2019.12.020; Notes: Notes: The “b coefficient” refers to the annual change within a linear regression model and is set in percent values. Different population structures apply to countries with varying income levels; so, cross-location comparisons are not valid for the metric YLD Counts.

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
