# Peer review of "Physical Rehabilitation Needs in the BRICS Nations from 1990 to 2017: Cross-National Analyses Using Data from the Global Burden of Disease Study"

_ijerph, 2020, doi:10.3390/ijerph17114139_

Round 1

Reviewer 1 Report

Thank you very much for providing me the opportunity to review this interesting manuscript

I congratulate the authors for their effort and the exhaustiveness of the presentation of their findings

Even though merely descriptive, the paper raises interesting points, in particular by underlining differences among countries with respect to their current (and probably future) needs in terms of physical rehabilitation

I do not have any specific comment

Perhaps however, the authors could just had a mention about how their research topic may be affected by the impact of COVID-19 pandemic in the near future, either regarding resources to be devoted or novel burden to appear

Of course, they cannot estimate this yet they could just cite it as a matter of caution for the future

Author Response

We submit our response through the attached.

Reviewer 2 Report

REVIEW Manuscript nr ijerph-820043

The quality of the manuscript is consistent and adds new evidence on the impact of Physical rehabilitation needs, in a long term perspective (1990-2017). In particular, the main hypothesis supported in this work is that in countries with an emerging economy (BRIC), the physical  rehabilitation needs  shows varied, overall significant statistical results. The design overall is appropriate for Int. J. Environ. Res. Public Health and the research question proposed should be of value and of sufficient interest to readers. . The bibliography seems consistent and detailed for the purpose of this study.  The limitations are accurate, in particular, the authors made it clear that the data used are “proxy data”, consequently, it is a search to be replicated with more accurate data.

However, there are some points needed to be further addressed before publication, generally minor concern issues.

  • The authors have not considered the hypothesis of extending the analysis on specific age groups for example elderly age groups?
  • Legend Table 1: *Conditions that may have early onsets and hence rather require pediatric physical rehabilitation. Zika virus can led to both neurological and musculoskeletal sequalae and physical rehabilitation interventions. ** When leading to long term impairments, pediatric conditions may also require adult physical rehabilitation services. In this legend the difference between * and two ** asterisks for Pediatric is not clear.

  • The authors used ANOVA to conclude their analysis. Have you done the normal test (Shapiro-Wilk) and the most important one of the Levene test? Specify. In addition, it would be good to provide statistics on the Bonferroni test (adjusted Bonferroni (1/5 of 0.05) in the post-hoc analysis, see Table 2.

  • It is not clear precisely which factor variables have been used (Income, Country and/or others? Etc). Is it possible to better specify the contents of ANOVA?

  • This is my question, have the authors considered comparing BRIC data with at least one developed country, such as the USA or a G7 country (see Introduction) as a case-control?

  • Second question (it is not a criticism): in order to make the analysis more complex one could not describe the trend of life expectancy for each of the five countries, at least the last ten years?

  • Third question: have the authors, not considered the hypothesis of integrating the analysis with a more formal trend analysis?

Author Response

(The authors gave the same response as above.)
